# Perceptions of Job Insecurity in Switzerland: Evidence Using Verbal and Numerical Descriptors

**DOI:** 10.3390/ijerph16101785

**Published:** 2019-05-20

**Authors:** Moreno Baruffini

**Affiliations:** Istituto di ricerche economiche (IRE), Università della Svizzera italiana (USI), Via Maderno 24—CP 4361, CH-6904 Lugano, Switzerland; moreno.baruffini@usi.ch; Tel.: +41-58-666-4116

**Keywords:** perceptions, job insecurity, flexibility, local labor markets

## Abstract

The purpose of this paper is to examine perceptions of job insecurity among employees, applying a panel model that allows us to account for the business cycle. In addition, the data will enable the comparison of two measures of job insecurity, one with a cardinal scale, and one with an ordinal scale. First of all, this paper carries out a descriptive analysis of job insecurity, taking into consideration two empirical measures, and using a panel data set from the longitudinal Swiss Household Panel (SHP). Second, an ordered probability unit (probit) model is applied, analyzing both workers’ concerns about job loss, and their subjective job security. Controlling for differences in socio-demographic and job characteristics, estimations show that perceptions of job security affect workers heterogeneously. This study contributes to the literature by revising variables that help to explain the formation of job insecurity perceptions.

## 1. Introduction

Studies conducted at the European level found that the performance of the worker is deeply influenced by the perception of job insecurity that alters their physical and psychological well-being [1,2], as well as the workers’ job satisfaction and general employment loyalty [3]. The perception of job insecurity may also demotivate workers to invest their time and financial resources in education and training, as opposed to the acquisition of additional firm-specific human capital [4]. In addition, the perception of job insecurity can directly affect consumption and employment, having, also, a negative effect on the worker’s side in bargaining [5]. Moreover, labor market flexibility, job security and their mutual relations are a lively policy issue [6,7]: Investigating the perception of job insecurity is therefore important for several reasons.

In a recent paper, Helbling and Kanji [8] studied the effects of both employees’ subjective assessments of their job security, fixed-term contracts, and the interrelationships between these two concepts in their impact on trajectories of life satisfaction. Studying the criteria of efficiency and effectiveness to be applied to labor market policies, one cannot, therefore, fail to know in detail the regional labor market, the skills level and the degree of dynamism and security of labor markets.

For these reasons, this paper aims at making two contributions to the literature. First, the paper adds empirical evidence on job security providing indirect evidence of differences at a Swiss national level. Secondly, considering two different aspects of job insecurity over a period, the study allows tracking changes in the perception of workers, addressing the question of how the perception of job insecurity among Swiss workers changes according to differences in socio-demographic and job characteristics.

The empirical analysis relies on nationally representative data from the longitudinal Swiss Household Panel (SHP), examining perceptions of job insecurity among Swiss workers between 2007 and 2013, a period that registered a temporary negative effect on the labor market (concomitant with the global crisis in the financial markets).

According to Dickerson and Green [9] the concept of employment insecurity “refers to all forms of welfare-reducing uncertainty surrounding employment, encompassing uncertainty over the continuity of the current job (job insecurity), uncertainty over the work itself, and uncertainty over future labor market prospects (employment insecurity)”.

At the empirical level, the theoretical and extensive definition is frequently simplified, taking into consideration only the probability of job loss or job insecurity, as measured in a qualitative form. These two aspects (probability of job loss or the degree of satisfaction with job security) depict indeed the mean expected loss arising from the uncertainty.

Following the literature, two measures to assess worker perceptions of job insecurity are therefore employed: One reveals workers’ concerns about job loss, and the second evaluates their degree of satisfaction with their job security using a verbal scale.

Switzerland provides a particularly appropriate market to examine the potential effects of “security” type arrangements due to the relatively high incidence of part-time contracts and flexible employment contracts: In 2010, 21.9% of the 182,000 temporary employees had a contract that would have expired within six months [10]. Another 53.5%, however, had an employment relationship in duration from six months to two years.

For these reasons, it is interesting to highlight the experience of Switzerland, which is a federal republic consisting of 26 regions (cantons). The geographical position in the European context, the federalist structure and the high share of immigrant workers are, indeed, remarkable factors that have led to a large regional variation, concerning financial capacity, employment patterns, unemployment levels and welfare dependency [11]. Switzerland is characterized by a relatively strong market performance with high rates of employment; federal laws regulate the policy of the labor market, but the canton is the level of implementation of concrete initiatives [12]. In general, the southwestern (French- and Italian-speaking) cantons show less advantageous labor market outcomes than their German-speaking counterparts [10].

In the last years, the Swiss labor market has performed above the Organization for Economic Co-operation and Development (OECD) average [13], especially considering its high employment rate, its exceptionally low unemployment, and its high wage levels. This has led, over the past three decades, to a steady workforce growth (from about 3 to 4.5 million people), coupled with high labor productivity. The main factors underlying this good performance are normally considered to be “a high degree of labor market flexibility, with decentralized wage bargaining, and relatively low employment protection regulations, supported by a strong focus (at least since the mid-1990s) on active labor market policies, and employment services characterized by strong ‘mutual-obligation’ principles” [12]. Another feature of labor market performance has been the design of immigration policies, which in the past “implied the use of immigrant labor as a labor supply reserve that left the country in times of economic hardship” [14]. Moreover, according to [15] “immigration created additional jobs in Switzerland by raising local demand for goods and, most importantly, services”. The transit from an industrial to a service economy implied an important change of the sector of employment of the working population in the industrial sector indeed. Even if the employment rate in the service sector rose considerably (from 39% to 73.7% in 2011), compared to the other European States [10], Switzerland has still a high employment ratio in different sectors, and a high level of employment in the manufacturing branches.

Nevertheless, Switzerland has not escaped the global economic crisis, which started in 2008: GDP growth revolved negative in the second half of the year, and remained negative in 2009; economic growth then resumed strongly in 2010 [16]. Unemployment increased by 30%, from 3.5% in 2008 to 4.4% in 2009, and only at the middle of 2013 did it start to recover (4.1% in the last quarter) [16]. The impact of the economic downturn on the labor market, at first relatively modest, was therefore substantial.

According to Eurofound [6] on the one hand, the fear of losing jobs is significantly lower in Switzerland than in the neighboring European countries, and the same holds for job satisfaction. The percentage on the actively occupied saying that they are “concerned or very concerned” about their job security, or “satisfied or very satisfied” with their job was higher in Switzerland than in the European Union both in 2005 and in 2010.

On the other hand, the working time and pressure term in Switzerland were above average, and there were more reports of bullying at work. Moreover, between 2005 and 2010, an increase was observed in two stress factors, namely, work and long hours of work under deadline pressure, and nowhere in the EU did these factors affect such, as in Switzerland.

Concerning our sample of Swiss workers, this study aims at assessing what are the main personal and job determinants of the perceptions of job insecurity among employees, and whether or not employees in flexible type works, or employed by fixed-term contracts, are more worried about job insecurity than those on permanent contracts.

Results show the importance of personal and job characteristics on the perception of job security, and they confirm that employees in flexible type works, or employed by fixed-term contracts, are more worried about job insecurity than those on permanent contracts.

## 2. Theoretical Framework

Over the last couple of decades, because of the intensification of competitiveness due to market globalization, together with the spread of new Information and Communication Technologies (ICT), policymakers have been trying to enhance the flexibility and improve the performance of European labor markets through the application of extensive labor market reforms. An aspect of these reforms has been the simplification of the restrictions regulating the use of temporary employment contracts [17], and the wide spreading of so-called flexible contracts, such as the fixed-term contract and temporary agency work. After the first period of an increasing feeling of insecurity, during the 1990s it decreased, although with important differences between countries [18].

Economic restructuring constitutes indeed a long-term phenomenon [19,20], driven by a global marketization, a greater service industry growth relative to growth in other sectors, and a rising premium for specialized skills related to technology shifts. Long-term work contracts are perceived to be costly, so their numbers are reduced, and all the factors outlined above foster employers’ perceptions of the need for market efficiency, which rationalizes a reduction of the workforce.

Consequently, full-time and long-term employees express insecurity, related to retaining current jobs and acquiring new ones; moreover, although mature workers may not “feel” the market forces that may be behind labor market changes, they perceive the relative insecurity of their own current employment. Are these concerns about job security related to the restructuring of their own occupations and industries? According to literature, workers in industrial sectors with high displacement rates, and in occupational sectors with high contingent employment rates, are more likely to perceive job insecurity [4].

Moreover, in principle, temporary employment can have both positive and negative welfare consequences for workers. A flexible scheduling arrangement and other aspects of the daily work experience related to temporary work, indeed, may be valued and preferred by some employees, whereas the insecurity and poorer working conditions associated with these contract types can have a negative impact on workers’ welfare [21]. Employment stability is instead desirable both for workers, who rank it as one of the most important factors for job satisfaction [22], and for firms, which dislike high turnover, and prefer stable employment relationships in order to retain human capital investment, and reduce both workforce screening and selection costs. On the other hand, the recent intensification of competitive pressures has called for more flexibility in labor markets for both firms and workers. Indeed, a growing literature shows that, in order to determine what dynamics impact on general workers’ well-being, the relationships between job satisfaction and job insecurity as economic variables is a crucial factor [23].

The main important issue regarding this area of interest is, therefore, to determine what factors influence the perception of security in the workplace, and its impact on the welfare of workers. Moreover, the economic literature [4,23] identifies two different relationships between job security and flexibility: A “rigid setting”, which implies a negative relationship between flexibility and security (a high level of job security can only be achieved at the cost of poor flexibility, and flexible employment patterns are in conflict with job security), and the “flexicurity” approach, which instead assumes that flexibility and security are not contradictions, but they can be mutually supportive, with the implementation of the right labor market policy.

We therefore formulate our expectations in the form of a research question, which will be particularly analyzed considering temporary employment.
Research Question 1a: “What are the main personal and job determinants of the perceptions of job insecurity among employees”?Research Question 1b: “Are employees in flexible type works, or employed by fixed-term contracts, any more worried about job insecurity than those on permanent contracts”?

This research aims at offering further insights into the topic of the relationship between perceptions of job insecurity, using a rich panel data, and taking into account the use of different measures of job insecurity, as previously discussed by Dickerson and Green [9].

Aiming at drawing significant empirical conclusions about the determinants of job insecurity, it is necessary to define and appropriately measure the concept of “insecurity”. As already presented in Section 1, “Job insecurity” is commonly intended to convey the probability that a worker will lose his current job, and then not have a comparable position [9,24,25]. Theoretical labor economists tend not to use this broad expression in their formal analysis of labor markets, but it is not clear what the term means in the theory of job search [26,27].

Empirical labor economists have generally avoided direct elicitation of worker expectations. At first, they have tried to figure out the perception of job insecurity from the statistics on unemployment rates and durations [28,29]. Nevertheless, reliable inference on expectations of achievements is difficult to achieve [30], because data on the labor market available to the researcher must be rich enough to allow the simulation of the process of formation of the presumed expectation. Moreover, a researcher must somehow know what information workers possess, and how to use this information to form expectations.

Direct elicitation of expectations is an alternative [9,25,31]. In consequence, survey questions on job security are generally posed in two major formulas. Commonly, individuals are required to indicate their degree of satisfaction with their job security by means of an ordinal verbal scale.

However, since the meaning of the verbal descriptors can vary between respondents, this formulation contains an important subjective component that makes the interpretation of the resulting measurement of the job insecurity problematic. Moreover, taking in consideration both the probability of job loss and the cost of job loss, the formulation complicates the perception of the respondent in at least two different components of the task.

Literature (as discussed by Dickerson and Green [9]) consequently suggests the use of a probabilistic question, which is to ask individuals about the probability of losing their jobs, as a common alternative to the above formulation. Dickerson and Green [9] have furthermore demonstrated the higher predictive power of such probabilistic questions and about individuals’ abilities to provide useful information in their responses to questions regarding their expectations of future job loss. Moreover, as soon as marginal changes in probability are proportionate along the scale, cardinal scales offer analytical advantages. This is not true of ordinal verbal descriptors.

Finally, according to existing literature [24,31,32], verbal descriptors can suffer from interpretation biases if their understanding of language is heterogeneous, or if the words are vague, and the meaning of the scale might differ among respondents, while the meaning of numerical scale points is unambiguous. In the analysis of expectations and realizations, it is consequently better to use cardinal rather than ordinal scales.

## 3. Data

The individual-level analysis, covering all the Swiss Cantons, has been realized using the data collected by the Swiss Household Panel (SHP), which is based at the Swiss Centre of Expertise in the Social Sciences (FORS). The Swiss National Science Foundation finances this project.

The SHP is an annual panel study based on a random sample of private households in Switzerland over time, interviewing all household members mainly by telephone, and the interdisciplinary and longitudinal study is well suited for representative analyses of the Swiss residential population. Data collection started in 1999, and in addition to the traditional variables found in national household surveys (income, health, housing and demographic characteristics), the SHP contains a series of questions on personal relationships and non-working actions (The data from the Swiss Household Panel is freely accessible to the scientific community on FORSbase (https://forscenter.ch/projects/swiss-household-panel/data/). It clearly contains some satisfaction questions also, including questions on satisfaction with job (in)security. According to the literature reported in the previous Section, the analysis included both of the two procedures mentioned above of eliciting job insecurity perceptions [4,25,30]: The degree of satisfaction with job security (job ins), and the probability of losing one’s job (job loss).

In the SHP survey two questions deal with these procedures:“Would you say that your job is very secure, quite secure, a bit insecure or very insecure?” Perceptions of job insecurity, as measured by this SHP survey question, will be the first outcome variable: “job ins”.“How do you evaluate the risk of becoming personally unemployed in the next 12 months, if 0 means “no risk at all” and 10 “a real risk”?”

Perceptions of job loss, as measured by this SHP survey question, will be the second outcome variable: “job loss”.

In the first case, respondents were given a verbal scale, “very secure”, “quite secure”, “a bit insecure” or “very insecure”.

In the second case, respondents were given an 11-point Likert scale, where 0 reflected a positive assessment (confidence—‘no risk at all’) and 10 reflected a negative assessment (fear—‘a real risk’) (Appendix A). The variable is re-coded for the econometric analysis (Appendix B).

The sample was restricted to the workforce aged 15–65: The size of the resulting sample is 5056 individuals for 7 years (2007–2013). The panel is strongly balanced: Each observation contains the same time points, and each observation contains the same number of observations.

The SHP provides individual and job-specific control variables: The set of control variables was selected according to previous studies that have defined the determinants of job insecurity perceptions [5,24,25,33,34,35,36]. Individual-specific explanatory variables, according to our explicit hypotheses, are gender, which distinguishes between male and female employees; marital status, between married and non-married workers; the presence of children in the household; education, defined on three levels; nationality, between Swiss citizens and foreigners; and age, measured in years.

Concerning job-specific attributes, the explanatory variables include the wage, the type of employment, and a public service indicator. Full-time and part-time constitute a dummy variable set, such that each worker can be assigned to one of these categories at any point in time. Fixed-term and temporary work constitute a separate dummy set. These variables capture possible increases in job insecurity, due to an expansion of non-standard jobs in the market [37].

Summary statistics for the samples used in particular descriptive and regression analyses are provided in Table 1.

The female rate was 51%, and the mean age was 41 years, and 29% of these workers had attained a very low level of qualification, (compulsory school, elementary vocational training, etc.), while 51% of workers had an intermediate education (apprenticeship, full-time vocational school, etc.), and the remaining 20% were highly educated (university degree or above). Fourteen percent of workers were employed on a fix-term contract, and 48% were employed in a part-time job. The percentage of workers employed in the public sector was 39%.

To investigate the research question about the main personal and job determinants of the perceptions of job insecurity, economic insecurity was at first analyzed, examining unconditional effects. On average, only 10% of workers expressed concerns about their perceptions of job insecurity (job ins), both in 2008, 2011 and in 2013, while conversely, perceptions of job loss (job loss) was 17.9% in 2008 and 18.4% in 2011, and then dropped to 16.91% in 2013.

Considering gender differences in perceptions of job insecurity, data show a large gender difference considering perceptions of job loss (job loss) as an insecurity measure; moreover, workers with higher education are more likely to perceive their work as safer.

According to the literature [4,25,30], finally, the probability of losing one’s job (job loss) tends to increase with age up to the point when workers reach retirement age, and the it declines afterwards. In contrast, according to our data, it increases monotonically with age. Considering occupational differences, data confirm results found in previous studies [24,25]. Occupations associated with state sector employment have low perceptions of job insecurity, and the insecurity, as expected, declines over the course of the considered period. In fact, their jobs are more secure than a private job in a downturn period. Those least worried about job insecurity are nevertheless supervisors and managers.

Perception of job insecurity is positively correlated with regard to differences in the conditions of the local labor market: Workers living in areas with better labor market outcomes (i.e., North Western Switzerland) are less likely to feel insecure than workers living in other regions, especially in border regions. A remarkable evidence is that workers residing in Ticino showed in 2008 low levels of insecurity, while in 2011 and 2013 they were the most concerned workers in Switzerland, maybe because of a “displacement fear” [38]. In fact, in the considered period the number of economically active foreign nationals in Ticino reached 25% of the active population, the highest value in Switzerland [10].

Workers living in small towns had greater job security than workers living in metropolitan areas during the considered period.

## 4. Methods

We next deepen the research question, which relates to whether employees in flexible type works, or employed by fixed-term contracts, are more worried about job insecurity than those on permanent contracts.

In order to take into consideration the variation, over the business cycle, of different factors to perceptions of job insecurity, and according to the literature [39], two different types of analysis were conducted. First, as in Linz and Semykina [25], time estimation was performed distinctly for different time periods: The period of relative stability, before the declining of the economy (2008), the period of major economic uncertainty, during the economic downturn (2011) and the period after the recovery (2013). Then, given the ordinal nature of the dependent variables, exploiting the longitudinal nature of the data, a Panel Ordered Probability Unit (Probit) model was applied.

In order to account for the ordered categorical character of the dependent variables, ordered probit regressions were applied, using perceptions of job insecurity and perceptions of job loss, as measured by Swiss Household Panel (SHP) survey questions, as proxy measures of job insecurity.
y**_it_* = β_1_**IS***_it_* + β_2_**JS***_it_* + μ*_it_* + ε*_it_*(1)
where *i* denotes individuals *i*, *i* = 1…*N*, and *t* denotes time. The latent dependent variable y* is perceived job insecurity or job loss, IS contains individual-specific regressors, and JS refers to job-specific regressors. The errors ε are assumed logistically distributed and independent across individuals and time, for given values of the regressors and the random intercept. The relationship between variables that vary at all levels is reflected by the structure of the model, which is commonly adopted by the previous literature on this topic [25,33,34,35,36,37,38,39].

## 5. Results

### 5.1. Cross-Sectional Ordered Probit Analysis

In the first step of the analysis, estimations were separately performed for 2008, 2011 and 2013, using, in the regression analysis, cross-sectional estimation techniques. For each dependent variable, the lowest value of the measure reflects the most favorable outcome (security-confidence), and the highest value reflects the most adverse outcome (insecurity-fear).

Table 2 presents the estimation results from Ordered Probit regressions of perceived job security on the set of controls that have been described in the previous section.

First, according to the literature [9,24,25], temporary workers should be most responsive in terms of job insecurity, as the risk of job loss mainly concerns temporary jobs. Effectively, for temporary workers, there is a sensible job security difference, being the coefficient for this category of workers, positive and strongly significant (row 16). Coefficients for 2011, both for perceptions of job loss (job loss) and perceptions of job insecurity (job ins) are lower than coefficients for 2008 and for 2013 (row 16, columns 2 and 5). This could be interpreted to reflect the generally less favorable conditions of the labor market in 2011 [10], and a general higher “feeling” of job insecurity that was also spread across between permanent workers.

#### 5.1.1. Age

Partly according to the descriptive analysis, but in contrast with what is often found in the analysis of subjective well-being measures [40,41], results do not clearly show that job security, expressed both as perceptions of job loss (job loss) and perceptions of job insecurity (job ins), is decreasing and convex (U-shaped) in age (rows 1–2).

#### 5.1.2. Education

Education seems to play a small but significant role in terms of perceptions of job insecurity (job ins). For 2011, the coefficient for job insecurity of low educated workers is positive and significant (row 10, column 5), reflecting the economic downturn, which mainly affected low-skilled jobs [10]. As discussed before, the fact that this formulation makes the interpretation of the resulting measurement of the job insecurity problematic, because it confuses the perception of the respondent in the probability of job loss and the cost of job loss, it must in any case be taken into consideration.

#### 5.1.3. Gender and Marital Status

The point estimate for married workers is negative, and significant only in 2008 and 2013 (row 4, columns 1–4–5). This implies that there is evidence that married workers felt less secure than single workers during the economic downturn.

As in previous studies [24], the presence of children in the household and cohabitation do not affect job security (row 5), both in case of married or non-married couples (row 6), before and during the recession, but the point estimate is statistically significant in 2011–2013. It shows that during and afterwards, a recovering period has an effect on perceived security.

Perceptions of job insecurity are lower among women in 2008 and 2013, and they are higher in 2011, but not significant (row 3, columns 4–6). These results are consistent with those observed in other European countries [4]: Usually, gender differences in job insecurity are slight.

When job insecurity is measured by perceptions of job loss, gender differences in perceptions are higher and significant for 2008 (row 3, column 1). This result is striking: One would expect, according to literature, that being a female should reduce one’s perceived job security. A possible interpretation here is that this is due to the high female participation rate in Switzerland [10].

#### 5.1.4. Region, Unemployment and Foreigners Rate

Regarding the effect of the region of residence, it is possible to see a clear relationship between the state of the labor market and the perception of job insecurity, after having controlled for other factors. The more the labor market is good (as in the central regions, as discussed in Section 1), the less job insecurity is perceived (rows 28–33, columns 1–6).

Moreover, immigrant workers feel more insecure than natives do (row 9, columns 1–6), as discussed by [34] and [24]. Nevertheless, the average local unemployment rate or foreign rate (rows 34–35, columns 1–6) should reduce perceived job security: The effect is mixed but insignificant. In 2011, therefore, the negative impact of the region of residence on job loss is rather high and significant at the 10% level (rows 28–33, column 2), indicating that during the period of substantial uncertainty the possibility of losing one’s job was equally perceived, regardless of the value of the unemployment rate.

Finally, concerning small towns, perceptions of job loss were relatively high in 2008, but decreased in 2011: Urban-rural differences diminished over time (row 12, columns 1–6).

### 5.2. Panel Order Probit

In the second step of the analysis, estimations were conducted on all the panel data set, using, in the regression analysis, panel data estimation techniques. As in the first step of the analysis, for each dependent variable, the lowest value of the measure reflects the most favorable outcome (security-confidence), and the highest value reflects the most adverse outcome (insecurity-fear).

Table 3 presents the estimation results from Panel Ordered Probit regressions of perceived job security on the set of controls that have been described in the previous paragraph. Year dummies were also included in the estimations in order to include aggregate trends.

The panel ordered probit regression mainly confirms general results, found in the first step of the analysis.

It confirms that temporary workers are the most responsive in terms of job insecurity as the risk of job loss mainly concerns temporary jobs (row 6, columns 1–2).

Concerning education (row 10–11, columns 1–2), the coefficients for job loss are statistically significant, positive for the lower level of education, and negative for the highest. This implies that, on the considered economic cycle, education showed a clear role in diminishing insecurity. Moreover, perceptions of job insecurity are lower among women (row 3, column 1–2), only considering job security, and in this case, results are in accordance to general literature [9,24,25].

Average local unemployment rate or foreign rate should reduce perceived job security: The coefficient is positive and significant only considering job loss (rows 40–41, column 1). Year dummies are significant only considering job loss (rows 34–39, columns 1–2), and, as expected, they are more significant for the years of the financial crisis.

### 5.3. Discussion

Results show that perceptions of Swiss workers are comparable to those obtained in other European economies applying the same methodologies. Perceptions of job security differ among workers according to differences in socio-demographic and job characteristics, affecting them heterogeneously.

In Europe, according to the Eurobarometer [6], workers felt more secure until 2008, and then the job insecurity started to rise in almost all European countries. Since then, with the progressive deterioration of macroeconomic conditions, workers’ views of their own job situation turned negative, and perceptions of job security are higher in Countries where employment opportunities are scarcer.

The considerable level of job insecurity of older workers in 2011 and 2013 (Table 2, row 8, columns 3–5–6) is consistent with the fact that, in the Swiss labor market, as in the US market, unemployment spells tended to be longer among them [42]. This is also similar to results reported in previous studies conducted in Europe [24].

Furthermore, the statistical non-significance of the average local unemployment rate or foreign rate on job security, versus the significant effect of the coefficient of the region of residence, found in the cross-sectional analysis (as discussed in Section 5.1), shows a simple correlation. Perceptions of the local unemployment are included in the general perception of the socio-demographic background of employees, described by the regional coefficient. Moreover, as discussed by Garz [37], ‘media effects’ (the general media coverage of labor market policy), could affect individual perceptions of job insecurity.

Finally, results seem to confirm the necessity to use cardinal rather than ordinal scales [9]. Verbal descriptors might differ among respondents if their understanding of language is heterogeneous, or if the words are vague, while the meaning of numerical scale points is unambiguous; cardinal scales are consequently preferable. Nevertheless, it is still an open question as to whether the responses on cardinal scales in practice can capture valid representations of what workers expect, given that not all respondents can show a perfect understanding of the questions [9].

However, the presence of heterogeneity, due to some individual and job characteristics in the potential sorting of workers, can imply some biases in the Panel Ordered Probit estimates. For this reason, estimation results have to be considered with caution. In any case, neither aspects may affect estimates of coefficients for aggregate unemployment rate or regional coefficients.

## 6. Conclusions

### 6.1. Main Evidences

The empirical investigation reported in this paper examined perceptions of economic insecurity in Switzerland, during the business cycle between 2008 and 2013, when a phase of severe economic downturn and recovery was registered. Our research question wanted to understand what the main personal and job determinants of the perceptions of job insecurity among employees are. Results are generally in line with previous results cited in the literature: Perceived job (in)security differs between workers with different age, education and marital status, and among workers living in different regions.

Nevertheless, Swiss data highlight that perceived job (in)security does not strongly differ between workers with different levels of education, and does not rigorously depend on age. This differs from previous studies.

Workers’ perceptions are consistent with actual labor market conditions: Due to the good performance of the Swiss labor market, the overall feeling of job insecurity appears to be less prevalent in Switzerland than in the rest of Europe. However, perceptions of economic security were very high in years of economic stability (until 2008) and deteriorated after the period of major uncertainty (2011). As expected, higher perceptions of job insecurity are common across temporary or part-time workers or non-standard jobs [25].

Confirming our prior expectations as formulated in the research question, employees in flexible type works, or employed by fixed-term contracts, were found to be more worried about job insecurity than those on permanent contracts were. As discussed by Berglund, Furåker, and Vulkan [43], the perceived risk of job loss “increases affective job insecurity, whereas both employment and income security have the opposite effect”. Moreover, “the effect of cognitive job insecurity on affective job insecurity is reduced in the presence of employment security, but is reinforced in the absence of it: Flexicurity may be a risky venture for employees”. A rise in perceptions of job security (measured as a declining job insecurity) over time among Swiss workers could imply a bias in the perceptions these workers make of their own job security.

### 6.2. Policy Implications

Results can be generalized to labor markets similar to the Swiss case, during a similar business cycle, due to megatrends (i.e., digitalization), or simple economic difficulties. Switzerland is a country characterized by innovative high-tech industry, with high levels of investment in research and development. Indeed, these are labor markets with a very low unemployment rate.

The findings have important implications from a theoretical perspective and a policy perspective. From a theoretical point of view, this study contributes to the literature by studying important variables that help to explain the formation of job insecurity perceptions. The findings support those who asked to emphasize the need to critically question rationality assumptions in many economic models, especially if these models involve perceptions or expectations [9]. For this reason, policymakers and managers should be aware that the risk of unemployment and job insecurity are perceived differently by different workers; otherwise, policies may have unanticipated economic consequences. Results show that in Switzerland, a “flexicurity” labor market approach seems to guarantee a general level of job security, but many regional differences affect the local job insecurity.

In this sense, further research is necessary to investigate other important aspects of the job security-perceptions relationship. In particular, it may be the case that the level of job protection legislation (EPL), and the level of job insecurity, are driving factors into the overall worker’s job satisfaction. Moreover, Green, Dickerson, Carruth and Campbell [34] claim that individuals with a history of unemployment, and those holding short-term employment contracts, are found to report the greatest levels of insecurity, concluding that that subjective measures provide useful information that may be used in further analyses of the workings of the labor market. Finally, a deep analysis of personal “self-efficacy” [44], as represented by the variables concerning “personal social attitude”, constitutes scope for future research.

## Figures and Tables

**Table 1 ijerph-16-01785-t001:** Descriptive statistics.

Variable	Description	Mean	Std. Dev.
*Individual and local characteristics*			
age	Age (continuous)	41.12	22.39
age2	Squared age (continuous)	2191.76	1902.23
female	1 if female	0.51	0.50
married	1 if married	0.47	0.50
children	1 if not-married couple with children	0.02	0.13
marchildren	1 if married couple with children	0.48	0.50
young	1 if age < 31	0.48	0.50
middleaged	1 if age > 31 & age < 50	0.35	0.48
older	1 if age > 50	0.28	0.45
lang1	1 if language French	0.37	0.48
lang2	1 if language German	0.70	0.46
lang3	1 if language Italian	0.05	0.21
swiss	1 if Swiss citizen	0.89	0.32
eduinf	1 if primary education	0.29	0.45
edumid	1 if apprenticeship, full-time vocational school	0.51	0.50
edusup	1 if high school, university	0.20	0.40
small_town	1 if 1 if lives in small or middle sized town	0.08	0.28
large_town	1 if 1 if lives in large town	0.19	0.39
*Job characteristics*			
lowinc	1 if low income	0.36	0.48
midinc	1 if medium income	0.19	0.40
highinc	1 if high income	0.38	0.49
temporary	1 if temporary job	0.14	0.35
parttime	1 if part-time job	0.48	0.50
proftrain	1 if doing professional training	0.30	0.46
public	1 if public sector job	0.39	0.49
professional	1 if professionals	0.17	0.38
hightech	1 if higher supervisory/technicians	0.24	0.43
desk	1 if intermediate occupations	0.20	0.40
self	1 if self employed	0.07	0.25
lowtech	1 if lower supervisors and technicians	0.02	0.15
service	1 if lower sales and service	0.12	0.32
technical	1 if lower technical	0.10	0.30
routine	1 if routine job	0.09	0.29
lowhour	1 if low than 25 work hours/week	0.24	0.43
midhour	1 if between 25 work hours/week and 42 work hours/week	0.38	0.49
highhour	1 if more than 42 work hours/week	0.29	0.45
nightwork	1 if night work	0.12	0.32
satwork	1 if work on Saturday	0.47	0.50
stresswork	1 if stressful job	0.34	0.47
lowint	1 if low intensity job	0.24	0.43
midint	1 if medium intensity job	0.54	0.50
highint	1 if high intensity job	0.22	0.41
*Region of residence*			
r1	1 if Lake Geneva (VD, VS, GE)	0.18	0.38
r2	1 if Middleland (BE, FR, SO, NE, JU)	0.25	0.43
r3	1 if North-west Switzerland (BS, BL, AG)	0.14	0.35
r4	1 if Zurich	0.17	0.37
r5	1 if East Switzerland (GL, SH, AR, AI, SG, GR, TG)	0.13	0.34
r6	1 if Central Switzerland (LU, UR, SZ, OW, NW, ZG)	0.10	0.30
r7	1 if Ticino	0.04	0.19
*Personal social attitude*			
Progressive	1 if left	0.23	0.42
Neutral	1 if center	0.56	0.50
Conservative	1 if right	0.11	0.31
*Employment characteristics*			
noga1	1 if Agriculture, hunting, forestry	0.04	0.18
noga2	1 if Fishing and fish farming	0.00	0.01
noga3	1 if Mining and quarrying	0.00	0.02
noga4	1 if Manufacturing	0.14	0.35
noga5	1 if Electricity, gas and water supply	0.01	0.08
noga6	1 if Construction	0.05	0.21
noga7	1 if Wholesale, retail; repair motor vehicles, household goods	0.12	0.32
noga8	1 if Hotels and restaurants	0.03	0.16
noga9	1 if Transport, storage and communication	0.05	0.22
noga10	1 if Financial intermediation; insurance	0.06	0.24
noga11	1 if Real estate; renting; computer; research	0.13	0.33
noga12	1 if Public admin, national defense; compulsory social security	0.07	0.25
noga13	1 if Education	0.10	0.31
noga14	1 if Health and social work	0.15	0.35
noga15	1 if Other community, social and personal service activities	0.07	0.25
noga16	1 if Private households with employed persons	0.00	0.03
*Macroeconomic condition*			
For	Local foreign rate	8.27	10.93
Un	Local unemployment rate	3.11	1.17

**Table 2 ijerph-16-01785-t002:** Ordered probability unit (probit) estimates for degree of satisfaction with job security (job ins) and probability of losing job (job loss), by period.

	Concerned about a Chance of Losing Job (*job loss*)			Concerned about Job Security (*job ins*)		
	2008	2011	2013	2008	2011	2013
*Individual and local characteristics (Reference categories: Single male workers with no higher education, age > 31 & age < 50, rural area)*						
age	0.094 ***	0.090 ***	0.131 ***	0.129 ***	0.117 ***	0.105 ***
	[0.026]	[0.022]	[0.025]	[0.024]	[0.021]	[0.025]
squared age	−0.001 ***	−0.001 ***	−0.002 ***	−0.001 ***	−0.001 ***	−0.001 ***
	[0.000]	[0.000]	[0.000]	[0.000]	[0.000]	[0.000]
female	−0.116 *	0.054	−0.066	−0.099	0.028	−0.103
	[0.063]	[0.061]	[0.064]	[0.065]	[0.064]	[0.066]
married	−0.222 ***	−0.024	−0.188 ***	−0.174 ***	−0.066	−0.087
	[0.068]	[0.064]	[0.068]	[0.066]	[0.065]	[0.068]
not-married couple with children	−0.045	−0.210	−0.331 *	0.089	−0.330 *	−0.321 **
	[0.186]	[0.173]	[0.194]	[0.203]	[0.175]	[0.160]
married couple with children	0.053	0.030	0.021	0.069	0.005	−0.040
	[0.064]	[0.059]	[0.062]	[0.062]	[0.060]	[0.063]
age < 31	0.106	−0.056	0.203	0.240 *	−0.041	−0.041
	[0.126]	[0.125]	[0.125]	[0.129]	[0.131]	[0.133]
age > 50	0.153	0.095	0.270 ***	0.085	0.210 **	0.252 ***
	[0.104]	[0.088]	[0.095]	[0.095]	[0.088]	[0.097]
Swiss citizen	−0.134 *	−0.155 **	−0.234 ***	−0.154 *	−0.102	−0.239 ***
	[0.081]	[0.079]	[0.081]	[0.083]	[0.077]	[0.088]
primary education	0.116	0.103	−0.032	0.161	0.244 ***	0.097
	[0.105]	[0.100]	[0.114]	[0.102]	[0.091]	[0.112]
high school, university degree	−0.096	−0.061	−0.062	0.066	0.009	0.028
	[0.059]	[0.058]	[0.059]	[0.061]	[0.060]	[0.060]
lives in small or middle sized town	0.274 ***	0.066	−0.122	0.234 ***	0.077	−0.015
	[0.086]	[0.087]	[0.091]	[0.088]	[0.079]	[0.088]
lives in large town	0.011	0.023	0.107*	−0.014	0.075	0.106 *
	[0.062]	[0.060]	[0.060]	[0.061]	[0.059]	[0.062]
*Job characteristics (Reference categories: Routine job, medium income)*						
low income	0.363 ***	−0.027	0.103	0.208 ***	0.015	0.160 **
	[0.073]	[0.074]	[0.075]	[0.074]	[0.073]	[0.076]
high income	0.043	−0.159 **	−0.186 ***	−0.075	−0.088	−0.112
	[0.068]	[0.067]	[0.068]	[0.066]	[0.066]	[0.070]
temporary job	1.004 ***	0.595 ***	0.717 ***	0.959 ***	0.778 ***	0.781 ***
	[0.105]	[0.104]	[0.108]	[0.118]	[0.114]	[0.115]
part-time job	−0.141 **	−0.235 ***	−0.118 *	−0.108 *	−0.072	−0.064
	[0.065]	[0.065]	[0.066]	[0.066]	[0.066]	[0.068]
doing professional training	−0.005	−0.068	−0.064	−0.023	−0.065	−0.081 *
	[0.048]	[0.048]	[0.048]	[0.049]	[0.048]	[0.049]
public sector job	−0.469 ***	−0.455 ***	−0.392 ***	−0.338 ***	−0.427 ***	−0.392 ***
	[0.053]	[0.051]	[0.053]	[0.051]	[0.050]	[0.051]
professionals	0.211 *	0.013	0.049	0.134	0.124	0.142
	[0.111]	[0.107]	[0.111]	[0.113]	[0.104]	[0.118]
higher supervisory/technicians	0.273 ***	−0.119	−0.098	0.144	0.058	−0.007
	[0.102]	[0.099]	[0.105]	[0.104]	[0.093]	[0.112]
intermediate occupations	0.241 **	−0.018	0.017	0.093	0.086	0.045
	[0.102]	[0.097]	[0.104]	[0.103]	[0.092]	[0.111]
lower supervisors and technicians	−0.183	0.026	0.099	−0.129	0.022	0.045
	[0.210]	[0.160]	[0.186]	[0.184]	[0.157]	[0.179]
lower sales and service	0.269 **	−0.046	0.144	0.033	0.126	0.082
	[0.110]	[0.106]	[0.113]	[0.110]	[0.096]	[0.116]
lower technical	0.211 *	−0.103	−0.057	0.198 *	0.164	0.017
	[0.118]	[0.112]	[0.119]	[0.118]	[0.106]	[0.124]
*Personal social attitude (Reference categories: Neutral)*						
progressive	0.032	−0.039	−0.051	0.048	−0.054	−0.040
	[0.055]	[0.054]	[0.056]	[0.054]	[0.055]	[0.057]
conservative	−0.100	−0.102	−0.103	−0.021	−0.108	−0.180 **
	[0.098]	[0.087]	[0.085]	[0.093]	[0.080]	[0.081]
*Region of residence (Reference categories: Zurich region)*						
Lake Geneva (VD, VS, GE)	−0.041	−0.096	0.124	−0.316 **	−0.259 ***	−0.053
	[0.130]	[0.098]	[0.104]	[0.128]	[0.099]	[0.103]
Middleland (BE, FR, SO, NE, JU)	−0.081	−0.191 **	0.133	−0.135	−0.172 *	−0.069
	[0.103]	[0.093]	[0.089]	[0.104]	[0.095]	[0.092]
North-west Switzerland (BS, BL, AG)	−0.085	−0.247 ***	0.049	0.076	−0.026	0.161 *
	[0.084]	[0.081]	[0.082]	[0.084]	[0.081]	[0.085]
East Switzerland (GL, SH, AR, AI, SG, GR, TG)	−0.132	−0.285 ***	−0.002	−0.083	−0.135	−0.059
	[0.090]	[0.092]	[0.096]	[0.090]	[0.087]	[0.096]
Central Switzerland (LU, UR, SZ, OW, NW, ZG)	−0.085	−0.203 **	−0.065	−0.093	−0.012	−0.101
	[0.101]	[0.096]	[0.105]	[0.101]	[0.098]	[0.102]
Ticino	−0.238	−0.334 *	0.183	−0.433 **	−0.093	−0.093
	[0.215]	[0.171]	[0.185]	[0.211]	[0.166]	[0.189]
*Macroeconomic condition*						
Local foreign rate	−0.008	−0.272	0.656	0.001	0.382	0.102
	[0.013]	[0.994]	[0.904]	[0.013]	[1.014]	[0.927]
Local unemployment rate	0.078	0.062	−0.012	0.023	−0.002	−0.066
	[0.097]	[0.057]	[0.058]	[0.096]	[0.057]	[0.057]
Observations	2521	2684	2489	2528	2694	2491

Robust standard errors in brackets. *** *p* < 0.01, ** *p* < 0.05, * *p* < 0.1.

**Table 3 ijerph-16-01785-t003:** Panel Ordered Probit estimates for degree of satisfaction with job security (job ins) and probability of losing job (job loss).

	Concerned about a Chance of Losing Job (*job loss*)	Concerned about Job Security (*job ins*)
*Individual and local characteristics (Reference categories: Single male workers with no higher education, age > 31 & age < 50, rural area)*		
age	0.111 ***	0.141 ***
	[0.014]	[0.015]
squared age	−0.001 ***	−0.002 ***
	[0.000]	[0.000]
female	−0.073	−0.089 *
	[0.046]	[0.051]
not-married couple with children	−0.166 ***	−0.125 ***
	[0.045]	[0.048]
children	−0.168	−0.120
	[0.119]	[0.125]
married couple with children	0.036	−0.048
	[0.039]	[0.041]
age < 31	0.042	−0.055
	[0.070]	[0.081]
age > 50	0.113 **	0.149 ***
	[0.052]	[0.054]
Swiss citizen	−0.222 ***	−0.248 ***
	[0.061]	[0.065]
primary education	0.137 *	0.079
	[0.071]	[0.073]
high school, university degree	−0.147 ***	−0.061
	[0.044]	[0.048]
lives in small or middle sized town	0.031	0.038
	[0.068]	[0.067]
lives in large town	0.024	0.027
	[0.044]	[0.048]
*Job characteristics (Reference categories: Routine job, medium income)*		
low income	0.122 ***	0.109 **
	[0.044]	[0.046]
high income	−0.138 ***	−0.140 ***
	[0.039]	[0.041]
temporary job	0.936 ***	0.968 ***
	[0.064]	[0.070]
part-time job	−0.111 ***	−0.003
	[0.042]	[0.046]
doing professional training	−0.068 ***	−0.098 ***
	[0.025]	[0.026]
public sector job	−0.431 ***	−0.351 ***
	[0.035]	[0.037]
professionals	0.099	0.185 **
	[0.077]	[0.082]
higher supervisory/technicians	−0.055	0.037
	[0.073]	[0.075]
intermediate occupations	−0.001	0.051
	[0.072]	[0.075]
lower supervisors and technicians	−0.038	−0.021
	[0.114]	[0.117]
lower sales and service	0.158 **	0.140 *
	[0.077]	[0.081]
lower technical	0.041	0.192 **
	[0.085]	[0.088]
*Personal social attitude (Reference categories: Neutral)*		
progressive	−0.014	0.012
	[0.033]	[0.035]
conservative	−0.083 *	−0.083 *
	[0.050]	[0.049]
*Region of residence (Reference categories: Zurich region)*		
Lake Geneva (VD, VS, GE)	−0.031	−0.241 ***
	[0.078]	[0.079]
Middleland (BE, FR, SO, NE, JU)	−0.002	−0.150 **
	[0.057]	[0.061]
North-west Switzerland (BS, BL, AG)	−0.090	0.026
	[0.063]	[0.070]
East Switzerland (GL, SH, AR, AI, SG, GR, TG)	−0.130 *	−0.137 *
	[0.071]	[0.075]
Central Switzerland (LU, UR, SZ, OW, NW, ZG)	−0.127 *	−0.052
	[0.075]	[0.079]
Ticino	−0.236	−0.211
	[0.145]	[0.134]
*Year effects*		
year2	0.132 ***	0.091 **
	[0.041]	[0.041]
year3	0.189 ***	0.127 ***
	[0.048]	[0.049]
year4	0.087	0.092
	[0.093]	[0.099]
year5	0.200 **	0.061
	[0.086]	[0.091]
year6	0.227 **	0.063
	[0.088]	[0.093]
year7	0.168 *	0.078
	[0.090]	[0.095]
*Macroeconomic condition*		
Local foreign rate	0.001	0.001
	[0.004]	[0.004]
Local unemployment rate	0.085 ***	0.002
	[0.027]	[0.028]
Observations	17,665	17,665
Number of idpers	5066	5066

Robust standard errors in brackets. *** *p* < 0.01, ** *p* < 0.05, * *p* < 0.1.

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
