# Peer review of "Perceptions of Job Insecurity in Switzerland: Evidence Using Verbal and Numerical Descriptors"

_ijerph, 2019, doi:10.3390/ijerph16101785_

Round 1

Reviewer 1 Report

The paper analyses determinants of job insecurity in Switzerland making use of rich panel data. The results indicate that job insecurity is heterogenously distributed across the workforce.

The paper is interesting, although the added value is comparatively low and extends an relatively crowded literature. Contributions to the existing literature should be presented more convincingly. Below you find some specific comments that might improve the paper.

Major Comments

-          The contribution to the existing literature apart from using Swiss data could be made clearer in the introduction, when deriving the research questions (especially w.r.t. the paper of Dickerson and Green 2012 (Ref. 9).

-          Although the title of the manuscripts suggests that analyses distinguish subjective and objective measures, it applies (in my opinion) two different subjective assessments. I would suggest to rephrase this more cautiously, not labelling it an “objective” measure or using an alternative measure indicating “objective job insecurity” if that even exists (an objective measure might be, e.g. some restructuring variables whether or not staff has been reduced in the respondent’s company in the past).

- Also, the abstract/title promises learning about the relevance of the business cycle. Although the analyses try to account for this, the discussion on this is a little thin, given that it is taken as one of the contributions made by the paper.

-          The discussion on the different measures of job insecurity is very detailed (p. 5). This could be streamlined and probably shifted to the data section when discussing the variables on job insecurity applied in the analyses.  A detailed discussion on the measurement of job insecurity can be found in Dickerson and Green 2012 (Ref. 9) and the author might refer to this detailed discussion instead.

-          The tables are poorly prepared. Instead of using the variable names from the analyses, where the reader has to scroll back and forth to table 1, you should choose the variable description so that the tables speak for themselves and probably work with subheadings for the different sets/constructs of control variables if necessary. It would be easier to read the table including additional rows for the respective reference categories.

-          Table 1 and 2 are very complex and confusing. I would suggest to skip Table 2 (or at least put it in the online appendix) as the added value as compared to the multivariate analyses presented later on is low and keep a well-prepared and meaningful table on the sample statistics including the mean and standard deviation.

-          Why is the political party included as control variable? I would suggest skipping this from the analyses or at least providing a convincing explanation.

-          The description of the estimated model should be moved to a “Methods” section. Moreover, the author should highlight why this model is chosen and refer to other studies applying a similar approach, as in l. 433 the author states “Results show that perceptions of Swiss workers are comparable to those obtained in other 433 European economies applying same methodologies”

-          If I understand it correctly, research question number 1 and 2 are jointly explored in the analyses. In a strict sense, they could be collapsed into one research question, given that the type of contract is also a job-related or personal determinant. In order to explore the 2. Research question more in-depth it would be interesting to perform separate analyses by type of contract.

-          Unobserved heterogeneity is arguably a big issue when analyzing job insecurity. The author is somehow silent about this issue, stating “Then, given the ordinal nature of the dependent variables, in order to control for unobserved heterogeneity and exploiting the longitudinal nature of the data, a Panel Ordered Probit model was applied. “ (p. 12, l.323). I think the author should elaborate on this issue: what does the applied method add? Would it be possible to perform the analyses controlling for time-constant unobserved heterogeneity (fixed effects), at least in a robustness check? Moreover this issue including the potential biases and the limits of interpretation of the results (no causal claims) should be discussed (e.g. in the discussion section). In this context, the wording should be partly chosen more cautiously (e.g. avoid talking about “effects”, “influences” etc.).

-          Given that the analyses are based on panel data I wonder whether a variable whether or not the respondent has been unemployed in the past could be included as control-variable in the analyses?

Minor comments

-          Chapter 2 is called “Methods” this term doesn’t fit that well. Terms such as “Theoretical considerations” or “Hypotheses” might fit the content better.

-          The Figures (1 and 2) are of poor quality and do not contain as much information justifying the space they need. I suggest skipping these figures and reporting the numbers in the text instead.

-          Some parts of the manuscript read in a fragmented way, when bullet points are used (p.5, data section). I’d prefer to revise this and write full sentences.

-          Any reference for the data for further information?

-          Line 210: I would prefer to name the Author names instead of introducing the sentence with reference [9]

-          The discussion of the results (esp. l. 410-430) could be condensed highlighting only to the main results and differences as compared to the cross-sectional model as it reads a bit choppy.

Author Response

Response to Reviewer 1 Comments

General comment: the paper analyses determinants of job insecurity in Switzerland making use of rich panel data. The results indicate that job insecurity is heterogenously distributed across the workforce.

The paper is interesting, although the added value is comparatively low and extends an relatively crowded literature. Contributions to the existing literature should be presented more convincingly. Below you find some specific comments that might improve the paper.

Response to the General Comment: I would like to thank the reviewer for the positive and challenging comments; I have tried to solve all the critical points, please find below all my responses to your comments.

Moreover, some sentences here and there in the paper have been modified in order to avoid repetitions. A native speaker at USI in Lugano proofread the paper.

Major Comments:

Major Comment 1: The contribution to the existing literature apart from using Swiss data could be made clearer in the introduction, when deriving the research questions (especially w.r.t. the paper of Dickerson and Green 2012 (Ref. 9).

Response Major Comment 1: I got the point: I have already written in the introduction section the two main contributions to the existing literature (lines 36-41), and, as suggested, I have rewritten the paragraph concerning the Research Question, highlighting as suggested the new perspective of my investigation.

Major Comment 2: Although the title of the manuscripts suggests that analyses distinguish subjective and objective measures, it applies (in my opinion) two different subjective assessments. I would suggest to rephrase this more cautiously, not labelling it an “objective” measure or using an alternative measure indicating “objective job insecurity” if that even exists (an objective measure might be, e.g. some restructuring variables whether or not staff has been reduced in the respondent’s company in the past).

Response Major Comment 2: Thanks for this comment: I have corrected the title, which now reads “Perceptions of Job Insecurity in Switzerland: Evidence using Verbal and Numerical Descriptors". My previous title was misleading because, as you noticed, I applied two different subjective assessments.

Major Comment 3: Also, the abstract/title promises learning about the relevance of the business cycle. Although the analyses try to account for this, the discussion on this is a little thin, given that it is taken as one of the contributions made by the paper.

Response Major Comment 3: Some sentences in the paper have been rewritten, in order to simplify the general presentation of results and I have extended the discussion on this topic in the conclusion section. I have also highlighted some issues for future research.

Major Comment 4: The discussion on the different measures of job insecurity is very detailed (p. 5). This could be streamlined and probably shifted to the data section when discussing the variables on job insecurity applied in the analyses.  A detailed discussion on the measurement of job insecurity can be found in Dickerson and Green 2012 (Ref. 9) and the author might refer to this detailed discussion instead.

Response Major Comment 4: I got the point: I have rewritten and shortened the paragraph of the “Theoretical framework” section, referring to the detailed discussion that can be found in Dickerson and Green 2012 (Ref. 9) (see also Response Minor Comment 1 and Response Major Comment 8).

Major Comment 5: The tables are poorly prepared. Instead of using the variable names from the analyses, where the reader has to scroll back and forth to table 1, you should choose the variable description so that the tables speak for themselves and probably work with subheadings for the different sets/constructs of control variables if necessary. It would be easier to read the table including additional rows for the respective reference categories.

Response Major Comment 5: I would like to thank the reviewer for both the above comments. I have re-formatted the tables, inserting variable descriptions, and additional rows containing subheadings and reference categories, in order to enhance the general clarity.

Major Comment 6: Table 1 and 2 are very complex and confusing. I would suggest to skip Table 2 (or at least put it in the online appendix) as the added value as compared to the multivariate analyses presented later on is low and keep a well-prepared and meaningful table on the sample statistics including the mean and standard deviation.

Response Major Comment 6: I got the point: I have skipped Table 2 and kept Table 1, which already contains variable descriptions, and additional rows containing subheadings, in order to enhance the general clarity.

Major Comment 7: Why is the political party included as control variable? I would suggest skipping this from the analyses or at least providing a convincing explanation.

Response Major Comment 7: I would like to thank the reviewer for this comment. I have checked the SHP questionnaires and I have modified the variables that now read: “conservative” “neutral” and “progressive”. These can be considered as proxies of personal “self-efficacy”, which can have a psychological effect on general perceived job security. Moreover, Switzerland has a direct democracy system, many citizens have a healthy personal social attitude, and these variables can affect their perceptions. I have also highlighted some issues for future research in the conclusion section.

Major Comment 8: The description of the estimated model should be moved to a “Methods” section. Moreover, the author should highlight why this model is chosen and refer to other studies applying a similar approach, as in l. 433 the author states “Results show that perceptions of Swiss workers are comparable to those obtained in other 433 European economies applying same methodologies”

Response Major Comment 8: I would like to thank the reviewer for this comment. I have added a “Methods” section that contains the description of the estimated model, where I have also highlighted why this model is chosen, according to the literature on the topic.

 Major Comment 9: If I understand it correctly, research question number 1 and 2 are jointly explored in the analyses. In a strict sense, they could be collapsed into one research question, given that the type of contract is also a job-related or personal determinant. In order to explore the 2. Research question more in-depth it would be interesting to perform separate analyses by type of contract.

Response Major Comment 9: I got the point: I have collapsed the two previous research question in a single question, highlighting as suggested the new perspective of my investigation (see also Response Major Comment 1). I have also highlighted some issues for future research concerning separate analyses by type of contract.

Major Comment 10: Unobserved heterogeneity is arguably a big issue when analyzing job insecurity. The author is somehow silent about this issue, stating “Then, given the ordinal nature of the dependent variables, in order to control for unobserved heterogeneity and exploiting the longitudinal nature of the data, a Panel Ordered Probit model was applied. “ (p. 12, l.323). I think the author should elaborate on this issue: what does the applied method add? Would it be possible to perform the analyses controlling for time-constant unobserved heterogeneity (fixed effects), at least in a robustness check? Moreover this issue including the potential biases and the limits of interpretation of the results (no causal claims) should be discussed (e.g. in the discussion section). In this context, the wording should be partly chosen more cautiously (e.g. avoid talking about “effects”, “influences” etc.).

Response Major Comment 10: I would like to thank the reviewer for this comment. Although a panel data model allows analysing observations that are not independent, the method cannot completely control for unobserved heterogeneity. Unfortunately, probit models with fixed effects lead to biased results (it does not exist a sufficient statistic allowing the fixed effects to be conditioned out of the likelihood). For this reason, I followed your suggestion, and I removed all the specific causal claims, selecting a more precise wording in order to avoid confusion; I have also inserted in the discussion section a final paragraph that claims for caution in interpreting those results.

Major Comment 11: Given that the analyses are based on panel data I wonder whether a variable whether or not the respondent has been unemployed in the past could be included as control-variable in the analyses?

Response Major Comment 11: Thanks for this comment: unfortunately, the SHP is mainly designed for sociological analyses, and it does not contain a variable that describes past personal unemployment cycles (it would be interesting to know the past story for individuals who experimented the big crisis during the nineties, when unemployment rate was higher than 9%...). For that reason, I have already inserted a paragraph in the Conclusion section that reads: “Moreover, Green, Dickerson, Carruth and Campbell [34] claim that individuals with a history of unemployment and those holding short-term employment contracts are found to report the greatest levels of insecurity, concluding that subjective measures provide useful information that may be used in further analyses of the workings of the labour market”.  This is therefore left for future research.

Minor comments:

Minor Comment 1: Chapter 2 is called “Methods” this term doesn’t fit that well. Terms such as “Theoretical considerations” or “Hypotheses” might fit the content better.

Response Minor Comment 1: Thanks for this comment: I got the point. I have corrected the section title, which now reads “Theoretical framework". I have inserted the following section, called “Methods”, where I have described the model (see also Response Major Comment 8).

Minor Comment 2: The Figures (1 and 2) are of poor quality and do not contain as much information justifying the space they need. I suggest skipping these figures and reporting the numbers in the text instead.

Response Minor Comment 2: Thanks for this comment: I got the point. I skipped the figures and reported the numbers in the text.

Minor Comment 3: Some parts of the manuscript read in a fragmented way, when bullet points are used (p.5, data section). I’d prefer to revise this and write full sentences.

Response Minor Comment 3: I got the point. I have deleted bullet points in that section, and I wrote full sentences.

Minor Comment 4: Any reference for the data for further information?

Response Minor Comment 4: If I have understood correctly, I have inserted a footnote (page 5) that contains more information about the SHP.

Minor Comment 5: Line 210: I would prefer to name the Author names instead of introducing the sentence with reference [9]

Response Minor Comment 5: I got the point. I have inserted Author names. I apologise for the problems that affected the previous version of the paper.

Minor Comment 5: The discussion of the results (esp. l. 410-430) could be condensed highlighting only to the main results and differences as compared to the cross-sectional model as it reads a bit choppy.

Response Minor Comment 6: I got the point. I removed all the repetitions through the paragraph, in order to enhance the clarity of the discussion (see also Response Major Comment 10).

Reviewer 2 Report

The topic of the paper is interesting, but serious concerns must be taken into account. 

In general terms:

Introduction:

Objective/Gap: the need for this research is not clearly stated. The main objectives of the study are not clear. The main research questions are not well developed.

you have considered data from longitudinal Swiss Household Panel (SHP), which examines perceptions of job insecurity among Swiss workers between 2008 and 2013. Why you did not consider data from the last 5 years OR from 2008 to 2018? Data is too old for this study. I have a serious concern in this regard.

Line 31: You should mention the author name before reference,...[8] studied the effects of both employees’. 

Theoretical development: The paper mentions “trade-off” theory, however, the theoretical assumptions of the research questions development are confusing and are not well justified.

The authors should explain why is better to focus their study on Switzerland country.

Theoretical Framework is missing is in manuscript.

How you have selected verbal scale?  

Line 243: “How do you evaluate the risk of becoming personally unemployed in the next 12 months, if 0 means “no risk at all” and 10 “a real risk”?” I will suggest attaching an appendix in the manuscript.

Why you have only considered Panel Ordered Probit regressions?  Any specific reason for this technique

Line 224: The data have been collected since 1999 through annually repeated surveys of household (This statement is confusing for readers).

Line 259: The sample was restricted to the workforce aged 15- 65, On which parameters you decided workforce age?

The text of the paper should be reviewed and corrected at the quotations references.

Also, the abbreviations should be detailed: SHP, OECD etc.

Although I think the topic raised in this paper is interesting, the paper should be improved substantively before considering a publication. First, as an empirical study, it fails to follow the standard process. Second, you'd better discuss the theoretical gap in the introduction section, and make your study based on solid theoretical foundations. Third, provide sufficient information on the measurement of the key variables, Four, discuss the potential contributions made to the literature and the implications for management before the end. This paper does not significantly contribute to the existing literature and it does not make a sufficiently novel contribution. 

Good luck with future work! 

Author Response

Response to Reviewer 2 Comments

General comment: The topic of the paper is interesting, but serious concerns must be taken into account.

Response to the General Comment: I would like to thank the reviewer for the challenging comments; I have tried to solve all the critical points, please find below all my responses to your comments.

Moreover, some sentences here and there in the paper have been modified to avoid repetitions. A native speaker at USI in Lugano proofread the paper.

Point 1: In general terms:

Introduction:

Objective/Gap: the need for this research is not clearly stated. The main objectives of the study are not clear. The main research questions are not well developed.

Response to Point 1: Thanks for this comment. I have already written in the introduction section the two main contributions to the existing literature (lines 36-41), and I have therefore rewritten the paragraph concerning the research question, highlighting as suggested the new perspective of my investigation. Moreover, I have corrected the title, which now reads “Perceptions of Job Insecurity in Switzerland: Evidence using Verbal and Numerical Descriptors”. (My previous title was misleading because I applied two different subjective assessments). Finally, some sentences in the paper have been rewritten and I have extended the discussion in the conclusion section. I have also highlighted some issues for future research. I apologise for the flaws in the previous version of the paper; some sentences have been modified to avoid repetitions or to correct quotation references.

Point 2: you have considered data from longitudinal Swiss Household Panel (SHP), which examines perceptions of job insecurity among Swiss workers between 2008 and 2013. Why you did not consider data from the last 5 years OR from 2008 to 2018? Data is too old for this study. I have a serious concern in this regard.

Response to Point 2: I have examined perceptions of job insecurity among Swiss workers between 2008 and 2013 because I wanted to account for the business cycle. As I wrote in the introduction section, Switzerland has not escaped the big global economic crisis, which started in 2008 and fully recovered (in Switzerland) only in 2013. I have extended the discussion on this topic in the conclusion section, and I have also highlighted some issues for future research.

Point 3: Line 31: You should mention the author name before reference,...[8] studied the effects of both employees’. 

Response to Point 3: I got the point. I have inserted Author names. I apologise for the problems that affected the previous version of the paper.

Point 4: Theoretical development: The paper mentions “trade-off” theory, however, the theoretical assumptions of the research questions development are confusing and are not well justified.

Response to Point 4: I would like to thank the reviewer for the comment. I have reconsidered all the previous literature, and I have modified the paragraph that now reads: “The main important issue, regarding this area of interest, is, therefore, to determine what factors influence the perception of security in the workplace and its impact on the welfare of workers. Moreover, the economic literature [4, 23] identifies two different relationships between job security and flexibility: a “rigid setting”, which implies a negative relationship between flexibility and security (a high level of job security can only be achieved at the cost of poor flexibility, and flexible employment patterns are in conflict with job security) and the “flexicurity” approach, which instead assumes that flexibility and security are not contradictions, but they can be mutually supportive, with the implementation of the right labour market policy.”

Moreover, I have collapsed the two previous research questions in a single question highlighting the new perspective of my investigation.

Point 5: The authors should explain why is better to focus their study on Switzerland country.

Response to Point 5: I have described in the introduction the peculiarities of the Swiss labour market, which is characterised by a relatively high incidence of part-time contracts and flexible employment contracts and show a high internal heterogeneity. I have rewritten some sentence to avoid repetitions and to enhance clarity. I have also extended the discussion on this topic in the conclusion section from a policy point of view.

Point 6: Theoretical Framework is missing is in manuscript.

Response to Point 6: Thanks for this comment: I have inserted the paragraph of the “Theoretical framework” section, referring to the detailed discussion that can be found in the literature.

Point 7: How you have selected verbal scale?

Response to Point 7: The verbal scale was defined following other European Institutions, by the Swiss Centre of Expertise in the Social Sciences (FORS), which provided the database that was used in this study.

Point 8: Line 243: “How do you evaluate the risk of becoming personally unemployed in the next 12 months, if 0 means “no risk at all” and 10 “a real risk”?” I will suggest attaching an appendix in the manuscript.

Response to Point 8: I would like to thank the reviewer for the above comment. I have shifted that part in Appendix.

Point 9: Why you have only considered Panel Ordered Probit regressions?  Any specific reason for this technique

Response to Point 9: I would like to thank the reviewer for this comment. I have added a “Methods” section, which contains the description of the estimated model, where I have also highlighted why this model is chosen, according to the literature on the topic.

Point 10: Line 224: The data have been collected since 1999 through annually repeated surveys of household (This statement is confusing for readers).

Response to Point 10: I got the point. I have rewritten the paragraph that now reads: “The individual-level analysis, covering all the Swiss Cantons, has been realised using the data collected by the Swiss Household Panel (SHP), which is based at the Swiss Centre of Expertise in the Social Sciences (FORS). The Swiss National Science Foundation finances this project. The SHP is an annual panel study based on a random sample of private households in Switzerland over time, interviewing all household members mainly by telephone, and the interdisciplinary and longitudinal study is well suited for representative analyses of the Swiss residential population. Data collection started in 1999 and in addition to the traditional variables found in national household surveys (income, health, housing and demographic characteristics), the SHP contains a series of questions on personal relationships and non-working actions”.

Point 11: Line 259: The sample was restricted to the workforce aged 15- 65, On which parameters you decided workforce age?

Response to Point 11: I decided working age on the base of the previous literature, which usually takes into consideration a "general working age". In Switzerland it is typically 15–65 years, because there is a rather high prevalence of the so-called “dual vocational education system”, which combines apprenticeship hours in a firm with school learning, and the general retirement age is 65 years.

Point 12: The text of the paper should be reviewed and corrected at the quotations references. Also, the abbreviations should be detailed: SHP, OECD etc.

Response to Point 12: I got the point. I have corrected quotation references, and I have detailed abbreviations. I apologise again for the problems that affected the previous version of the paper.

Point 13: Although I think the topic raised in this paper is interesting, the paper should be improved substantively before considering a publication. First, as an empirical study, it fails to follow the standard process. Second, you'd better discuss the theoretical gap in the introduction section, and make your study based on solid theoretical foundations. Third, provide sufficient information on the measurement of the key variables, Four, discuss the potential contributions made to the literature and the implications for management before the end. This paper does not significantly contribute to the existing literature and it does not make a sufficiently novel contribution.

Response to Point 13: Thanks again for the challenging comments; as I have written at the beginning, I have tried to solve all the critical points.

At first, I have corrected the second section title, which now reads “Theoretical framework”. I have already written in the introduction section the two main contributions to the existing literature, and I have rewritten the paragraph concerning the Research Question, highlighting as suggested the new perspective of my investigation (I have collapsed the two previous Research Questions in a single question). I have rewritten and shortened the paragraph of the “Theoretical framework” section, referring to the detailed discussion that can be found in Dickerson and Green 2012 (Ref. 9). I have added a “Methods” section that contains the description of the estimated model, where I have also highlighted why this model is chosen, according to the literature on the topic. I have also highlighted some issues for future research concerning separate analyses by type of contract.

I have re-formatted the tables, inserting variable descriptions, and additional rows containing subheadings and reference categories, to enhance the general clarity.

Finally, some sentences in the paper have been rewritten, to simplify the general presentation of results; I have also extended the discussion on the topic in the conclusion section.

I am grateful for your insightful suggestions, which have led to significant changes in my manuscript. I apologise again for the problems that affected the previous version of the paper.

Good luck with future work! 

Thank you very much!

Round 2

Reviewer 1 Report

From my perspective, the author addresses all issues raised satisfactorily.

Author Response

Response to Reviewer 1 Comments

General comment: From my perspective, the author addresses all issues raised satisfactorily.

Response to the General Comment: I would like to thank the reviewer for the final positive comment!

I am grateful for your worthy suggestions, which have led to significant changes in my manuscript. I apologise again for the problems that affected the previous version of the paper.

Reviewer 2 Report

Introduction section should be revised. No need to mention paper structure (Line 110)

Line: In the second case, respondents were given an 11-point Likert scale, You need to give a reference for adopting the 11-point Likert scale. (Attach Questionnaire if possible)

The conclusion section is too long. You can divide it into Policy implications. 

Author Response

Response to Reviewer 2 Comments

Point 1: Introduction section should be revised. No need to mention paper structure (Line 110).

Response to Point 1: Thanks for this comment. I have revised the introduction, removing the paper structure paragraph.

Point 2: Line: In the second case, respondents were given an 11-point Likert scale, You need to give a reference for adopting the 11-point Likert scale. (Attach Questionnaire if possible)

Response to Point 2: I got the point: I have inserted a second Appendix reporting the part of the Questionnaire with the question and the complete Likert scale.

Point 3: The conclusion section is too long. You can divide it into Policy implications

Response to Point 3: Thanks for this comment: as suggested, I have revised the final section, introducing a subsection titled "Policy Implications".

I would like to thank the reviewer for all the comments; I have tried to solve all the points, thanks again.
